# Are You “Nudgeable”? Factors Affecting the Acceptance of Healthy Eating Nudges in a Cafeteria Setting

**DOI:** 10.3390/ijerph19074107

**Published:** 2022-03-30

**Authors:** Christine Kawa, Wim H. Gijselaers, Jan F. H. Nijhuis, Patrizia M. Ianiro-Dahm

**Affiliations:** 1Department of Management Sciences, University of Applied Sciences Bonn-Rhein-Sieg, 53359 Rheinbach, Germany; patrizia.ianirodahm@h-brs.de; 2Department of Educational Research and Development, School of Business and Economics, Maastricht University, 6211 LM Maastricht, The Netherlands; w.gijselaers@maastrichtuniversity.nl (W.H.G.); jfh.nijhuis@maastrichtuniversity.nl (J.F.H.N.)

**Keywords:** nudge, healthy eating, acceptance, cafeteria, health intervention, susceptibility, nudgeability

## Abstract

Research has identified nudging as a promising and effective tool to improve healthy eating behavior in a cafeteria setting. However, it remains unclear who is and who is not “nudgeable” (susceptible to nudges). An important influencing factor at the individual level is nudge acceptance. While some progress has been made in determining influences on the acceptance of healthy eating nudges, research on how personal characteristics (such as the perception of social norms) affect nudge acceptance remains scarce. We conducted a survey on 1032 university students to assess the acceptance of nine different types of healthy eating nudges in a cafeteria setting with four influential factors (social norms, health-promoting collaboration, responsibility to promote healthy eating, and procrastination). These factors are likely to play a role within a university and a cafeteria setting. The present study showed that key influential factors of nudge acceptance were the perceived responsibility to promote healthy eating and health-promoting collaboration. We also identified three different student clusters with respect to nudge acceptance, demonstrating that not all nudges were accepted equally. In particular, default, salience, and priming nudges were at least moderately accepted regardless of the degree of nudgeability. Our findings provide useful policy implications for nudge development by university, cafeteria, and public health officials. Recommendations are formulated for strengthening the theoretical background of nudge acceptance and the susceptibility to nudges.

## 1. Introduction

The interest in nudging as a health intervention to improve eating behaviors in individuals is increasing. About 70% (440 reports) of empirical evidence in the field of nudging has focused on dietary behavior [1]. Nudges change the decision-making context of an individual such that their behavior becomes more predictable. They facilitate choosing a certain preferred option by changing the environment in which the decision is made. Nudges neither change the incentives for making a specific decision nor forbid or remove a less preferred option. Thus, nudges can be conceived as changes in the environment that can take many different forms. They can be applied by any person responsible for creating a decision-making context (called choice architects) [2]. The dual-process theory of the mind can be considered as a theoretical background of nudging [2,3]. Nudges can affect either our automatic information processing system (called system 1) or our deliberative system (called system 2). When using system 1, we make fast, intuitive, and automatic decisions. When using system 2, our decisions are slower, deliberate, and controlled [2,3]. Positive effects of nudges on eating behavior have been found across the entire life span and in various settings [4]. Reports from 36 reviews and empirical studies showed positive results in 80% of these studies [5]. Others showed that nudging is a promising tool to increase healthy eating behaviors, specifically in cafeteria settings [6,7]. For example, nudging is associated with increased fruit consumption (*r* = 0.43) [8] or choosing healthy food options in 22% of university students (*n* = 104) visiting a cafeteria [9].

Despite the reported effectiveness of nudges on food consumption, many studies on nudging as an intervention to increase vegetable consumption in school-like settings showed weak effects or were of moderate methodological quality [8,10]. Other researchers considered the effect sizes of nudges promoting healthy eating to be small (Cohen’s *d* = 0.23) [11] or nudges to result in multi-directional effects [12]. Nudge effectiveness also depends on the type of nudge [13,14], suggesting that one-nudge-fits-all designs will fail to achieve population-broad objectives. For example, nudges applied in a cafeteria improved the eating behavior in employees but resulted in the opposite behavior in students. These mixed findings were attributed to differences in nudge acceptance. It was hypothesized that students did not accept the applied nudges as often as the employees did, resulting in more unhealthy food choices and yielding the nudge ineffective [15]. In this sense, nudge acceptance can be viewed as tolerating the nudge’s influence on one’s behavior. In summary, it remains unclear under which conditions nudges work and for whom they work [12,16,17]. Nevertheless, a higher nudge acceptance is associated with increased nudge effectiveness. As a consequence, researchers have called for more research on assessing the acceptance of nudge interventions to gain more knowledge on the working mechanisms of a nudge and determine what makes a nudge effective [17].

In general, a majority of people accept healthy eating nudges [18] and thus tolerate the influence a nudge might have on their behavior. Most individuals also want to follow a healthy diet [19]. Between 57–71% (depending on the type of nudge) of a representative sample from the UK accepted healthy eating nudges [20]. A representative German sample was described as being open to healthy eating nudges, with 51% of the respondents rating the concept of nudging as (very) positive. Specific nudge interventions, targeted at the general population, were accepted by 71% of the general German population [21]. Default nudges (e.g., pre-selecting a favorable option) were strongly accepted when applied in a cafeteria to promote healthy eating [22]. Research has shown that nudge acceptance is associated with nudge effectiveness [23]. The variance in nudge acceptance, depending on the type of nudge, showed that certain nudges can be perceived as discomforting, manipulative, or coercive—thus being less acceptable [24]. Without nudge acceptance, behavioral change is less likely to occur [25]. Hagman proposed a nudge acceptance model in which nudge acceptance is conceptualized as the conscious rating of a nudge as an intervention. In this model, several factors influence nudge acceptance, for example, the nudge technique (type of nudge) [25]. This means that the underlying working mechanisms of a nudge (whether system 1 or system 2 is activated by the nudge) can influence nudge acceptance. Nudge acceptance depends on several properties of a nudge: (1) whether a nudge biases a person to engage in a specific behavior or debiases a person from a behavior (bias or debias), (2) whether an individual can deduce the intention of the nudge as well as the means of behavioral change (transparency), (3) whether a nudge has societal motives or engages in cognitive or emotional information processing (social aspects), (4) whether a nudge will cause friction when implemented (ease of use). In this kind of research, nudge acceptance is directly linked to nudge effectiveness (actual behavioral change) [25]. We propose that a specific type of nudge that is highly accepted will be highly effective. In addition to the influential factors on nudge acceptance proposed by Hagman [25], several other influences on nudge acceptance have been assessed so far: culture [23,26], political opinions [27,28], individual preferences and habits [29,30], disclosure of the nudge’s purpose [31,32], and psychological factors such as personality traits [28]. Still, other influential factors such as personal characteristics may exist [33,34]. Consequently, we argue that other influential factors on nudge acceptance exist as well.

The environment in which a person makes a decision is by definition especially important in nudging. Choice architects are the individuals who create the decision-making context in a specific setting [2]. Thus, they have the opportunity to construct a specific environment in which nudges as well as certain factors promoting health behavior can be applied. A university cafeteria is an environment in which students meet, gather, eat together, and socialize. In this setting, choice architects have the opportunity to create a context in which, for example, social norms of eating healthily are salient in order to foster healthy eating. Social norms provide guidelines on what kind of behavior is expected by other individuals in a certain situation [35]. According to Hagman’s nudge acceptance model, the social aspects of nudging affect nudge acceptance [25]. Research has shown that perceiving the strong social norms of healthy eating imposed by one’s immediate surroundings increased the acceptance of healthy eating nudges in a cafeteria [36]. Because social norms provide guidelines for appropriate behavior, nudges employing the social norms of healthy eating as a working mechanism were found to influence and improve food choices in a systematic review study [35]. Despite this positive finding, other studies on these social norm nudges found mixed results. A recent quantitative review found 12 studies on social norm nudges within the health context. This review stated that in 29% of the reviewed cases social norm nudges effectively improved health behavior [12]. Based on reviews and empirical studies, we propose that nudges based on social norms have the potential to influence nudge acceptance as well as an individual’s eating behavior, but more research is needed. We tested the following hypothesis: (H1) perceived social norms of eating healthy have a positive influence on nudge acceptance.

Next to social norms, health-promoting collaboration is another possible influencing factor on nudge acceptance and effectiveness. Health-promoting collaboration is defined as an environment in which all individuals are highly committed to values involving and fostering health. All relevant individuals in this given environment share this commitment and collaborate on it [37]. The concept of health-promoting collaboration is in line with the principles of a Health Promoting University (HPU) laid out in the Okanagan Charter [38]. The aim of an HPU initiative is to incorporate health into the culture, policies, and processes of a university, for example by creating healthy environments and promoting well-being [38]. This focus on the environment makes the combination of health-promoting collaboration and nudging an interesting research topic. Thus far, health-promoting collaboration has not yet been researched in the context of nudging but has been found to be associated with paying attention to one’s own health in a workplace context [37]. In a university setting, choice architects can also apply this factor purposely. Both social norms and health-promotion collaboration focus on sharing beliefs and values on a specific topic in a given situation. While social norms are imposed externally by the immediate context in which a person acts, health-promoting collaboration focuses on collegial collaboration and shared norms and values. More explicitly, the influence of social norms stems from the external surroundings of an individual. Other individuals dictate what kind of behavior is appropriate in the given situation. Health-promoting collaboration is more about the internal beliefs and values of an individual that are shared with others in a given setting. It is also about reciprocal mindfulness and encouragement in order to promote health-oriented behavior in oneself and others [37]. Similar to social norms, health-promoting collaboration creates an environment in which the appropriate behavior is obvious: specifically, an environment in which health plays an important role. This environment is created by the university itself and also by other social groups existing within the university. Therefore, it is possible that health-promoting collaboration can vary within the same general setting. We propose that health-promoting collaboration is likely to affect the acceptance of healthy eating nudges. We tested the following hypothesis: (H2) health-promoting collaboration has a positive influence on nudge acceptance.

Since choice architects within a university and cafeteria setting create the environment in which food decisions are made, they also have a certain degree of responsibility to create an optimal environment for their students and customers. Individuals can differ in the degree to which they feel that a choice architect in a cafeteria and university is responsible or obligated to arrange the environment to promote healthy eating. This perceived responsibility or obligation of a cafeteria or university was found to increase nudge acceptance in a previous study [36]. The more a person perceived cafeteria and university officials as responsible for promoting healthy eating, the more this person accepted healthy eating nudges. Recent research in nudging has focused on the importance of transparency. Knowing the source of a nudge (and thus who applies it) was found to be important in nudge acceptance [25,39] and effectiveness [25,40]. We tested the following hypothesis: (H3) perceived responsibility of a university or cafeteria to promote healthy eating has a positive influence on nudge acceptance.

Thus far, individual psychological factors such as personality traits have rarely been assessed in the nudging context [28]. The research focus is often limited to aspects of the nudge itself or to the choice architects applying the nudge [41]. Individual psychological factors should be considered when designing nudges [34], for example, by making certain options in a decision-making context more accessible [2]. Certain psychological factors such as an individual’s autonomous motivation have been associated with nudge acceptance, revealing that individuals are more likely to accept nudges that target behavior for which they show autonomous motivation [41]. Other psychological factors such as procrastination have not yet been researched regarding nudge acceptance. Procrastination is the tendency to postpone unwanted or detested tasks to the furthest possible point in time instead of executing them immediately or in a timely manner. This behavior can occur in different aspects of an individual’s life. It often arises within a university context where it is expressed in postponing study-related tasks [42,43]. Procrastination is also prevalent when engaging in health-related behaviors, often resulting in negative consequences for the quality of the task and the person’s well-being [42]. Procrastination is one reason why nudges are necessary, because their effortless impact facilitates making certain preferred choices [2]. Therefore, nudges can counteract procrastinating behavior and prevent the negative effects of procrastination from occurring. Thus far, procrastination has been negatively associated with the intention to engage in health behaviors [44]. Furthermore, it was linked to poorer health in students [45] and poorer eating behavior such as dieting [43]. Individuals who procrastinate are likely to benefit from nudges and are more likely to accept nudges because they perceive them as helpful facilitators to circumvent their procrastination. We therefore propose the following hypothesis: (H4) a tendency to procrastinate has a positive influence on nudge acceptance.

Research has shown that nudge effectiveness and acceptance are influenced by certain factors. Most recently, attention has also been paid to the so-called equity effects of nudges. Equity effects in nudging mean that not all individuals benefit equally from one specific nudge [46]. Regarding nudge effectiveness, a recent systematic review of 38 studies found that healthy eating nudges have differential effects on different target groups. In this review, 30.4% of the nudges created a more positive outcome for individuals that were socioeconomically advantaged compared to individuals that were socioeconomically disadvantaged [47]. In addition, a specific nudge that had been effectively applied to improve eating behavior in middle-aged samples was ineffective in young adults [48]. A similar healthy eating nudge applied in an online cafeteria setting was effective for employees but not for students [49]. The cause of these differential effects is unclear. Possibly, they are caused by personal characteristics [33]. Research has also shown that the effectiveness of nudges varies in terms of their underlying mechanisms [41]. In nudge development, the underlying working mechanisms of nudges are often based on the so-called MINDSPACE framework [50]. This framework summarizes nine different working mechanisms: messenger (paying more attention to given information because its messenger is perceived as credible), incentives (providing information on the costs and benefits of certain behavior), norms (portraying socially accepted behavior in a given situation), default (providing a specific pre-selected choice), salience (making certain behavior more visible and salient), priming (making certain knowledge more accessible), affect (conveying a certain emotion), commitment (making an individual commit to something), and ego (conveying a positive and consistent self-image) [50]. Nudges that combined priming and salience effects as working mechanisms were found to be the most effective types of nudges in a systematic review on healthy food and beverage nudges [51]. These findings showed that a one-size-fits-all approach for nudging is ineffective [47,52]. However, the potential differential effects of nudging have not been fully researched [33]. Thorough knowledge of a nudge’s target group is helpful in nudge development. Personalizing nudges to meet the needs of the target group can increase nudge acceptance and, consequently, nudge effectiveness [24,28]. Nudge acceptance is linked to nudge effectiveness [23,25], thus making it more likely that a person is influenced by the nudge. To make nudges more effective, research on the susceptibility to nudges (nudgeability) and a systematic grouping of the influential factors of nudge effectiveness are called for [53]. To draw conclusions on nudgeability (susceptibility to nudges) regarding healthy eating nudges, we propose the following research questions: Can we identify groups of individuals that vary in their degree of nudge acceptance—and are thus more or less nudgeable? Do groups varying in their degree of nudgeability differ regarding influential factors such as social norms, health-promoting collaboration, responsibility, and procrastination?

The theoretical background on nudging has been criticized for being imprecise and for lacking a robust framework [54]. It also remains unclear under which conditions nudges work and for whom they work [12,16,17], while more research on assessing the acceptance of nudge interventions as well as nudgeability is called for [17,53]. It is important to investigate who is and who is not nudgeable, to determine which individuals will benefit from nudge implementation [41]. Certain influences on nudge acceptance have been researched, some with mixed results [35] and others that have not yet been assessed even though they should be considered given previous research findings [34]. Therefore, this study aims at providing more insight into the acceptance of different types of nudges and on finding factors that influence nudgeability. (1) We assessed the effect of several potential influencing factors on the acceptance of different types of healthy eating nudges: social norms, health-promoting collaboration, responsibility, and procrastination. These influencing factors are likely to arise in a university and cafeteria setting. (2) We aimed at gaining more knowledge on what makes a person more or less nudgeable by clustering individuals together. The clusters were based on individual nudge scores, and we compared them regarding several influential factors. To this end, we assessed nine different types of nudges to draw conclusions on which type of nudge is suitable for more or less nudgeable individuals. We drew practical implications for developing healthy eating nudges to be applied in a university and cafeteria. Identifying further influential factors on nudge acceptance that have not yet been researched or have shown mixed results in previous studies provides useful information for the theoretical background on nudging.

## 2. Materials and Methods

### 2.1. Study Design and Procedure

Data were collected from 4 October to 7 December 2021 using an online questionnaire. All (*N* = 9526) students at a German university of Applied Sciences were invited via e-mail to participate in a student health survey assessing their health status and possible influential factors on their health (convenience sampling).

All participants were informed about the purpose of the study and personal data security. They actively gave their consent in order to participate. The present study is part of a larger research project which was approved by the ethical committee of the University of Bonn (sequence number 086/19) and is in line with the ethical standards of the Declaration of Helsinki. This research project’s general aim is to assess the health and well-being of the university’s students and employees as well as to compose health interventions and to offer programs to improve the health and well-being of these individuals in different areas. As compensation, all participants had the option to be included in a lottery for various health-related products (such as a yoga mat).

### 2.2. Measures

#### 2.2.1. Nudge Acceptance

Thus far, no commonly used tool exists to measure nudge acceptance [21]. Nørnberg and colleagues [36] proposed a scale for assessing nudge acceptance that we adapted for the present study and translated into German (see Appendix A). They validated it using factor analysis and reliability analysis. The factor structure showed high eigenvalues and a very good Cronbach’s α of 0.848 for the overall nudge acceptance. The scale by Nørnberg and colleagues [36] was carefully developed based on the MINDSPACE framework often used in nudge development [36]. In this framework, 9 different underlying working mechanisms for nudging are described: messenger, incentives, norms, default, salience, priming, affect, commitment, and ego [50]. Nørnberg and colleagues utilized this framework to construct 10 items—each item assessing the acceptance of a different type of nudge. Their types of nudges each rely on the 9 working mechanisms proposed in the MINDSPACE framework, while the incentive working mechanism is represented in 2 nudges [36,50]. We adapted this nudge acceptance scale further by using the first eight types of nudges (messenger, incentive 1, incentive 2, norms, default, salience, priming, and affect) and added a ninth type (priming and salience). We added this ninth nudge because this type of nudge had been previously applied to the population of the present study in an online experiment [49]. In addition, a combination of priming and salience effects in nudging were found to be very effective [51]. The original English scale as well as our German translation are shown in Appendix A. For each individual nudge, participants were asked to indicate their level of agreement to different healthy eating interventions in a university’s cafeteria on a 5-point scale (from agree to do not agree). Higher values indicated less nudge acceptance. Each healthy eating intervention represented a different type of nudge. To prevent any order effects, we randomized the order in which the nudges were presented to the participants. No instructions for answering the items on nudge acceptance were provided by Nørnberg and colleagues [36]. Because Germans are not very familiar with the concept of nudging [21], we described the nudges as healthy eating interventions (Appendix A).

#### 2.2.2. Factor Influencing Nudge Acceptance

Social norms. We asked the participants to indicate the degree to which their own vegetable intake was influenced by friends and family (social norms) using three items [36]. Participants answered on a 5-point scale (from agree to do not agree). Lower values indicated a higher level of perceived social norms. The items were formulated by Nørnberg and colleagues and showed an acceptable Cronbach’s α of 0.659 [36]. We translated these items into German. See Appendix A for the English original and the German translation.

Health-promoting collaboration. Health-promoting collaboration focuses on the role played by health within the university among colleagues regarding health perception and health relevance. This scale was developed based on the Health-Oriented Leadership Questionnaire [55] and translated into German [37]. Originally, it consisted of 11 items showing a good Cronbach’s α of 0.860. Participants answered on a 5-point scale (from does not apply at all to completely applies) [37]. For time–economic reasons, we included six items. Each item represented a subscale of health-promoting collaboration: (1) self-care value, (2) self-care awareness, (3) self-care behavior, (4) fellow-student-care value, (5) fellow-student-care awareness, and (6) fellow-student-care behavior. Self-care describes how individuals deal with their own health in respect to others. Fellow-student-care comprises how individuals deal with others regarding health aspects. High values in health-promoting collaboration mean that a strong collegial collaboration at the university exists, where students focus on values, awareness, and behavior surrounding the health and well-being of themselves as well as others [37].

Responsibility to promote healthy eating. We assessed the participants’ attitudes toward a cafeteria’s or university’s responsibility in promoting their health using the item: I think it is the school’s or the canteen’s obligation to try and improve my vegetable intake. Participants answered our German translation on a 5-point scale (from agree to do not agree); thus, lower values indicated a higher level of perceived responsibility. This item was formulated by Nørnberg and colleagues [36]. Originally, they assessed responsibility using two oppositely coded items (see Appendix A) with a good Cronbach’s α of 0.725.

Procrastination. Procrastination describes the behavior of completing unwanted, unpleasant, or even despised tasks later rather than in the present moment. We used the Prokrastinationsfragebogen für Studierende (PFS), which specifically focuses on student procrastination within a university setting. The PFS is validated with an acceptable reliability based on factor loadings. It was measured using four items on a 5-point scale (from almost never to almost always). Higher values represented a proneness to procrastinating study-related tasks within a university setting [42].

#### 2.2.3. Demographics

Participants indicated their gender as female, male or diverse. Age was assessed using the categories 17–24, 25–30, and 31 years or older. Participants were also asked to indicate the department of their studies (Computer Science; Electrical Engineering, Mechanical Engineering and Technical Journalism; Natural Sciences; Management Sciences; Social Policy; and Social Security Studies) and whether they were in a bachelor’s or master’s program.

### 2.3. Sample

The population of students at the university where the study took place consisted of *N* = 9256 students. Of these students, *n* = 1047 completed the questionnaire, yielding a response rate of 11%. After removing participants because of missing values, *n* = 1036 students remained. We further identified four outliers in the sample regarding the nudge acceptance. The final sample consisted of *n* = 1032 participants (60.3% females; 38.9% males; 0.8% diverse). Of the participants 71.8% were 17–24 years old, 22.0% were 25–30 years old, and 6.1% were 31 years or older. The participants were distributed among the different departments as follows: 14.4% in Computer Science, 20.2% in Electrical Engineering, Mechanical Engineering, and Technical Journalism, 17.9% in Natural Sciences, 37.3% in Management Sciences, and 10.3% in Social Policy and Social Security Studies. Of the participants, 83.3% were studying to receive a bachelor’s degree and 16.7% were studying to receive a master’s degree. Considering the population of students in the present study, our sample can be described as representative in terms of their pursuit of either a bachelor’s or a master’s degree. Compared to the overall population, our sample was slightly younger. More female than male students participated in the study, while the overall population consisted of more male than female students. The Computer Science Department was slightly under-represented in our sample. The assessment of representativeness was based on the university’s internal demographic statistics (internal unpublished document).

### 2.4. Data Analysis

We calculated an overall nudge acceptance mean based on the acceptance of the nine types of nudges. To test our hypotheses, we conducted multiple regression analyses on the overall nudge acceptance using forced entry (“enter” in terms of SPSS) as a method. In this way, all factors (social norms, health-promoting collaboration, responsibility, and procrastination) are forced into the model simultaneously not determining any order [56]. We also assessed the effect of the influential factors on each type of nudge in nine separate multiple regression analyses. 

To identify individuals that could be grouped together based on their individual nudge acceptance scores on each of the nine nudges, we conducted several cluster analyses following a recommended approach [56,57,58]. First, we prepared our data by recoding the nudge acceptance rating of each nudge, social norms, and responsibility (5 now indicating higher values). Second, to determine the number of clusters, we randomly split the sample into two groups and conducted a hierarchical cluster analysis applying Ward’s method for both groups separately [57]. As a similarity measure we used the squared Euclidean distance (SED), which is commonly used in the field of health psychology [58]. Third, we conducted a non-hierarchical cluster analysis (k-means algorithm, dictating the number of clusters), providing a more accurate cluster membership. To determine any difference for the influential factors between clusters we used multiple variate analyses and Bonferroni post hoc tests.

For all analyses, we used SPSS v. 28 (IBM SPSS Statistics for Windows, Version 28.0. Armonk, NY, USA: IBM Corp), applied a significance level of 0.05, and deleted missing values listwise.

## 3. Results

### 3.1. Descriptive Statistics

The overall nudge acceptance was moderate, while acceptance of the individual nudges varied. Default, priming, salience, and affect nudges showed a high acceptance rate. The acceptance rates for the messenger and incentive 2 nudges were moderate, while the norms nudge, the incentive 1 nudge, and the priming and salience nudge were less accepted. Standard deviations were large, indicating that individuals varied in their nudge acceptance. Table 1 summarizes the descriptive statistics for nudge acceptance and reports an acceptable Cronbach’s α reliability. The default, priming, and salience nudges showed rather high acceptance ratings. The affect and incentive 2 nudges had a moderate acceptance, while acceptance for the norms, incentive 1, and priming and salience nudges was low.

Social norms regarding healthy eating were perceived to be moderate. The same was true for the time pressure. The responsibility of the cafeteria or university to promote healthy eating was perceived as high. Participants indicated a moderate health status compared to their peers as well as moderate health-promoting collaboration. They rated healthy eating as important to them. Participants viewed themselves as self-efficient, while work engagement and procrastination were moderate. Table 2 summarizes the descriptive statistics for the influential factors and reports questionable to excellent Cronbach’s α reliability.

### 3.2. Influences on Nudge Acceptance

#### 3.2.1. The Influence of Social Norms, Health-Promoting Collaboration, Responsibility, and Procrastination on Overall Nudge Acceptance

To test our hypotheses, we conducted a multiple regression analysis on the overall nudge acceptance (Table 3). We proposed that social norms (H1), health-promoting collaboration (H2), responsibility (H3), and procrastination (H4) have a positive influence on nudge acceptance. All variance inflation factors (VIFs) were between 1.02 and 1.05, indicating no multicollinearity. Social norms did not have a significant influence on overall nudge acceptance, even though the *p*-value was very close to reaching significance. We reject hypothesis H1. Furthermore, the analysis revealed health-promoting collaboration and responsibility as significant influences on overall nudge acceptance. The significant positive influence of responsibility on overall nudge acceptance means that the more the participants perceived a cafeteria or university to be responsible for promoting health, the higher was the overall nudge acceptance. H2 can be confirmed. Lower values on the nudge acceptance scale indicate a higher nudge acceptance. Accordingly, the significant negative influence of health-promoting collaboration on overall nudge acceptance means that the more health-promoting collaboration the participants perceived, the more they accepted nudges. Therefore, H3 can be confirmed. Procrastination did not have an influence on overall nudge acceptance, and we reject hypothesis H4. The model explains 16.9% of the variance in overall nudge acceptance.

#### 3.2.2. The Influence of Social Norms, Health-Promoting Collaboration, Responsibility, and Procrastination on the Acceptance of Nine Different Types of Nudges

We performed separate multiple regression analyses on the acceptance of each type of nudge to gain more detailed information (Table 4). These analyses revealed the perceived responsibility (by the participants) of a cafeteria or university to promote healthy eating to be a significant and positive influential factor on the acceptance of each type of nudge. Health-promoting collaboration was revealed to be a mediocre influence on nudge acceptance. The stronger the health-promoting collaboration was, the more likely three out of nine nudges (incentive 2, default, and salience) were accepted. Social norms and procrastination were weak influencing factors on nudge acceptance, as they only significantly influenced the acceptance of one out of nine nudges (social norms influenced incentive 1 nudge; procrastination influenced priming and salience nudge). The variance of acceptance explained for each type of nudge ranged from 3.1% (priming and salience) to 9.2% (norms).

### 3.3. Identifying Groups of Nudgeable Individuals

To answer our research question of whether we can identify groups of individuals varying in their degree of nudge acceptance (nudgeability), we conducted a cluster analysis. First, we randomly split the sample into two subsamples to determine the number of groups. For each subsample, we executed a hierarchical cluster analysis using Ward’s method. In these analyses, only those who rated all nine types of nudges (*n* = 981) were included. Based on the squared Euclidean distance (SED) coefficients in the agglomeration schedule for both samples (subsample 1: *n* = 488, SED 5 = 4498.73, SED 4 = 4710.87, SED 3 = 4961.11, SED 2 = 5355.70, SED 1 = 6906.79; subsample 2: *n* = 493, SED 5 = 4685.22, SED 4 = 4918.15, SED 3 = 5289.91, SED 2 = 5745.16, SED 1 = 6951.66), we decided to set the number of groups at three.

Next, we performed a k-means non-hierarchical cluster analysis (20 iterations) to assign cases to the three groups. This analysis reached the three-cluster solution after 14 iterations. All nine nudge ratings contributed highly and significantly to the clustering process (all *p* < 0.001). The three groups were characterized as follows (Table 5). The first group was characterized by low scores on all types of nudges. It was especially unresponsive to the messenger, incentive 1, incentive 2, and norms nudges, as well as priming and salience nudges (*un-nudgeable*). The second group was characterized by high scores on all types of nudges, indicating a high responsiveness to all types (*nudgeable*), especially the default, salience, priming, and affect nudges. The third group was mixed in their nudge acceptance (*conditionally mixed nudgeable*). It included high scores for the same nudges as the high nudge acceptance group (default, salience and priming) and lower scores for the same nudges as the low nudge acceptance group (messenger, incentive 1, incentive 2, and norms).

Descriptive statistics for the three groups of nudgeable individuals are shown in Table 6. The un-nudgeable group was the smallest cluster and included slightly more female than male students. Most students were 17–24 years old, while only a few students were 31 years or older. The nudgeable group was the second largest cluster and included slightly more female than male students. The majority of students were 17–24 years old, while only a small percentage of students were 31 years or older. The conditionally mixed nudgeable group was the largest of the three clusters, with more female than male students. The largest proportion of students was 17–24 years old, while only a few students were 31 years or older.

We found significant differences between the three groups regarding social norms, health-promoting collaboration, and perceived responsibility (by the participants) of the university or cafeteria to promote healthy eating behavior (multivariate analyses in Table 7). Bonferroni post hoc tests further distinguished differences between the groups. The differences between the three groups for social norms and health-promoting collaboration were caused by a significant difference between the nudgeable and un-nudgeable groups. The nudgeable group perceived stronger social norms and a stronger health-promoting collaboration than the un-nudgeable group. All three groups differed significantly in their perceptions of the responsibility a university or cafeteria has to promote healthy eating. The nudgeable group perceived the cafeteria or university to be more responsible for promoting healthy eating than did the other two groups. The same was true when comparing the conditionally mixed nudgeable and un-nudgeable groups. The three groups did not differ in procrastination.

We concluded that students could be grouped into three different groups based on their nudge acceptance of the nine different types of nudges. While the nudgeable and un-nudgeable groups were united in their high or low acceptance ratings of the different nudges, the conditionally mixed nudgeable group was divided. This cluster accepted certain types of nudges, while others were less accepted. In differentiating further between these clusters, the factors of overall nudge acceptance, perceived responsibility for a university or cafeteria to promote healthy eating, social norms, and health-promoting collaboration were especially relevant. The nudgeable group scored the highest on these factors, while the un-nudgeable group scored the lowest. Procrastination did not play a role.

## 4. Discussion

This study had two specific aims: (1) explaining the acceptance of healthy eating nudges in a university cafeteria setting using different factors that were likely to arise there; (2) determining what makes a person more or less nudgeable (susceptible to nudges). We found overall nudge acceptance to be moderate among subjects and it varied from nudge to nudge. Acceptance of the default, priming, salience, and affect nudges was high. These acceptance levels can be expected based on previous findings on nudge effectiveness of these types of nudges [51]. The messenger and incentive 2 nudges were moderately accepted, while acceptance of the norms, incentive 1, as well as priming and salience nudges was low. Specifically, the low acceptance rate of the priming and salience nudge was unexpected, as such a combination of nudges was previously found to be effective [51]. The specific design and working mechanisms of a nudge thus played an important role in the acceptance of nudges, undermining a one-nudge-fits-all design. In particular, the responsibility to promote healthy eating as well as a health-promoting collaboration positively influenced nudge acceptance. We also identified three clusters of individuals with varying levels of nudge acceptance: the un-nudgeable group, the conditionally mixed nudgeable group, and the nudgeable group. These groups also differed in responsibility, social norms, and health-promoting collaboration, but not in procrastination.

### 4.1. The Influence of Social Norms, Health-Promoting Collaboration, Responsibility, and Procrastination on Nudge Acceptance

Research has previously shown that the perceived responsibility of a cafeteria or university to promote healthy eating increases nudge acceptance [36]. We not only found the same results for overall nudge acceptance but also when considering all nine nudges (messenger, incentive 1, incentive 2, norms, default, salience, priming, affect, and priming and salience) individually. The perceived responsibility of a cafeteria or university to promote healthy eating was the strongest influence on nudge acceptance in our study. It influenced the acceptance of all nine nudges and overall nudge acceptance with a high significance. The nine nudges differed in the specific working mechanism behind them. For example, the norms nudge compared one’s own healthy eating behavior to that of peers, while the default nudge automatically provided a salad as a side dish to any chosen main dish. Despite these differences, all nudges portrayed the university or cafeteria as the source of the nudge. Thus, the nudges were transparent because the participants knew who implemented them. Our results underlined recent research that found transparency in healthy eating nudges to be of particular importance in nudge acceptance [25,39] and effectiveness [25,40].

Health-promoting collaboration is a newly developed concept that has not yet been assessed in the context of nudge acceptance. Because of its similarities to social norms, we proposed that a strong health-promoting collaboration among peers positively influences nudge acceptance. We were able to confirm this proposition for the overall nudge acceptance as well as for three types of nudges (incentive 2, default, and salience nudge). Feeling supported by others in one’s own health and acting within an environment where individuals are highly committed to healthiness were likely to positively influence nudge acceptance. The three nudges for which health-promoting collaboration was a significant influence on nudge acceptance are suitable examples of such a collaboration. First, the incentive 2 nudge uses so-called scare campaigns to promote healthy eating by showing extreme pictures of disease. Accepting the portrayal of such pictures indicates strong and shared values regarding healthiness. Second, the default nudge automatically provides a salad side dish with every main dish. By accepting this nudge, an individual automatically agrees to this health-promoting collaboration. Third, the salience nudge provides salient information on healthy eating in the form of a poster. Such a poster indirectly applies a health-promoting collaboration. It increases the awareness of one’s own health, which is an important aspect of health-promoting collaboration. Healthy eating nudges are by definition health-promoting, because their intention is to foster healthy environments [2]. Thus far, universities employing HPU initiatives to improve the health and well-being of students put rather a lot of effort into shaping the university setting accordingly [38]. This effort can possibly be facilitated by utilizing nudges. 

Another factor that has not yet been researched in nudge acceptance is procrastination. This factor is very relevant, because it is one of the reasons why nudging is an important intervention. Nudges facilitate decisions that people who are prone to procrastination otherwise might not make [2]. We found procrastination to influence only the acceptance of the priming and salience nudge. Individuals prone to procrastinate accepted this specific nudge more. None of the nudges used in the present study specifically targeted the working mechanisms behind procrastination [36], and the sample in this study did not show a strong tendency to procrastinate. Therefore, procrastination did not seem to be very relevant in the current sample of students.

Previous findings showed that perceiving strong social norms of healthy eating within one’s immediate surroundings predicted a higher acceptance of healthy eating nudges [36]. In contrast, social norm nudges often show rather mixed findings [12,35]. We found social norms to influence only the incentive 1 nudge. This nudge applied to a healthy eating competition and could not be associated with social norms. Interestingly, social norms did not influence the norms nudge. Explaining these findings is difficult, but they fit into the category of unclear results regarding social norm nudges. Similarly, a recent systematic review focusing on nudging in the food waste area only found four high-quality studies on social norm nudging to be effective. This review mentions that social norm should be applied with care as interventions, because they might reinforce negative behavior [14].

### 4.2. Identifying Groups of Nudgeable Individuals

Little research has been conducted on so-called nudgeability and the systematic grouping of relevant individual features [53]. Thus far, we know that so-called equity effects exist, which means that nudges affect individuals in different ways. For example, some nudges targeting dietary behaviors (such as nudges that provide information) were found to be more beneficial to less socioeconomically disadvantaged people, while other nudges (such as nudges facilitating behavior) were more beneficial to more socioeconomically disadvantaged people [47]. We added to the existing knowledge on differential nudge effects by identifying three clusters of students and relating these clusters to other influential factors. This allowed us to describe groups of students that were either nudgeable, conditionally mixed nudgeable, or un-nudgeable.

In the nudgeable group of students, almost all nudges were equally highly accepted. An exception was the priming and salience nudge, which was only moderately accepted. These students perceived a cafeteria or university to be significantly more responsible for promoting healthy eating than did the conditionally mixed nudgeable and un-nudgeable groups. They also perceived a moderate health-promoting collaboration to be present. This perception was equally moderate in all three groups. The nudgeable group perceived more social norms than the un-nudgeable group, but not the conditionally mixed nudgeable group. Thus, nudgeable students seemed to highly accept healthy eating nudges, especially when applied by an institution (cafeteria or university). They also felt a stronger need for institutions to involve themselves in promoting healthy eating. We propose that this group of students will easily accept nudges applied by institutions.

Students of the conditionally mixed nudgeable group were divided in their acceptance. While accepting certain nudges strongly (default, salience, priming, and affect nudges), other nudges (messenger, incentive 1, incentive 2, norms, and priming and salience nudges) were not accepted. When considering this cluster of students in terms of comparison to the other two clusters on the influential factors, the conditionally mixed nudgeable students were similar to the nudgeable students. Both clusters portrayed equal and moderate values for the perceived social norms of eating healthily and health-promoting collaboration. Their difference lay in the overall nudge acceptance as well as in the perceived responsibility of a cafeteria or university to intervene. Both values were moderate and significantly lower for the conditionally mixed nudgeable students compared to the nudgeable students. For these students it seems that accepting a nudge depended on certain conditions. Perceiving social norms and health-promoting collaboration was important to them when it came to accepting nudges. In particular, the default (pre-selected healthy side dish) and salience nudges (poster with tips of healthy eating) could be interpreted to include aspects of social norms. The priming (prompting to buy healthy dishes) and affect nudges (healthy names for healthy dishes) could be interpreted to portray health-promoting collaboration. On the contrary, this group of students, for example, did not accept the incentive 1 nudge. In this nudge, a competition for healthy eating behavior was proposed. Driving someone to compete with others was not in line with their perceived importance of health-promoting collaboration. These students preferred certain nudges while disliking others, which was in line with the general description of this cluster. They could only be nudged by certain nudges—for example, the default, salience, priming, and affect nudges.

The un-nudgeable group of students showed low acceptance of six of nine nudges. They only moderately accepted the default, salience, and priming nudges. These students scored significantly lower on all influential factors (except health-promoting collaboration, compared to the high nudge acceptance students). They neither felt the need for the cafeteria to involve itself in healthy eating promotion nor valued involvement by family and peers or fellow students. They appeared generally uninterested.

The default, salience, and priming nudges appeared to be the most appropriate ones for application in a university and cafeteria setting for students. They were at least moderately accepted by students regardless of their degree of nudgeability. The names given to these three clusters already provide a vantage point for nudge development. Nudges developed by institutions for the nudgeable students should focus on identifying the institution providing the nudge, while nudge development for the conditionally mixed nudgeable students should predominantly focus on default, salience, priming, and affect nudges, which foster a health-promoting collaboration and portray social norms. Nudge development for un-nudgeable and uninterested students remains difficult. Neither family, friends, and peers, nor an institution, seemed to matter to this group in nudge acceptance. At the very least, they moderately accepted default, salience, and priming nudges. Nudge development for this group of students should therefore focus on these working mechanisms of nudges. Another possible source of nudging is the person him- or herself. Self-initiated nudges have been described as imposed by the individuals themselves by consciously and voluntarily changing their own choice architecture [59,60]. These self-applied nudges preserve an individual’s autonomy and independence and might therefore be specifically suitable for the un-nudgeable students.

### 4.3. Limitations and Future Research

In the present study, we only assessed the hypothetical nudge acceptance and linked it to nudge effectiveness using theory. Future research should directly connect the nudge acceptance and nudge effectiveness of healthy eating nudges and preferably in an experimental setting measuring actual nudge acceptance. In this way, more robust findings will contribute to understanding the link between nudge acceptance and nudge effectiveness. We only assessed a limited number of influential factors on nudge acceptance, while others are likely to exist. Research on further influential factors will improve our understanding of nudge acceptance and facilitate the development of effective nudges. The present study focused on the target group of university students. As nudges can have different effects on different target groups, other target groups should be considered in future studies as well. We found health-promoting collaboration to positively influence nudge acceptance. This newly developed concept needs to be researched more thoroughly in order to draw concrete practical implications from it. Designing and testing a nudge that purposefully enhances students’ perception of a health-promoting collaboration is in line with the HPU framework and may potentially be beneficial. While we were able to formulate clear descriptions of nudgeable students and conditionally mixed nudgeable students, the description of un-nudgeable students remains vague and undistinguished. More research is needed on systematically grouping individuals according to their nudge acceptance and effectiveness. It will also be interesting to research whether a similar cluster formation can be found in other samples. Furthermore, psychological factors (such as personality traits) may yield interesting results in grouping nudgeable individuals. Thus, we can focus on changing their status from un-nudgeable to nudgeable.

### 4.4. Practical and Theoretical Implications

Our study provides useful practical implications for nudge development, particularly for university and cafeteria officials, but also for public health officials. We identified the perceived responsibility of a cafeteria or university to promote healthy food consumption as positively influencing nudge acceptance. In our study, we made clear that the hypothetical interventions (nudges) to which the participants indicated their acceptance were implemented by university or cafeteria officials. Thus, participants were aware of the source of the nudges. Cafeterias and universities should take up this responsibility and transparently apply nudges to foster healthy eating behaviors. Note that these nudges must be tested rigorously before application. University officials in particular should foster health-promoting collaboration, as this factor also positively influences nudge acceptance. One way of doing so is by employing HPU initiatives within a university [38]. These initiatives themselves have a great potential to foster health-promoting collaboration within the university’s community, which in turn facilitates nudge acceptance and is likely to increase nudge effectiveness. Then, officials take responsibility for promoting healthy eating and create synergetic effects. Creating an environment in which all students value health and healthy eating in particular, as well as support each other in their shared values and beliefs, is likely to increase nudge acceptance and consequently nudge effectiveness. University, cafeteria, and public health officials have the opportunity to create these environments and implement nudges because they are choice architects. We identified three groups of students differing in their nudgeability. Students identified as nudgeable can easily be nudged by providing the source of the nudge (e.g., the institution). Conditionally mixed nudgeable students accept nudges such as default, salience, priming, and affect nudges, which portray healthy eating social norms and health-promoting collaboration. Un-nudgeable students are difficult to nudge, but our results show that default, salience, and priming nudges applied in the form of self-nudging are promising. Our findings suggest that not all individuals may benefit equally from nudges. This needs to be considered in nudge development.

Next to practical implications, we also provide useful information for the theoretical background on nudging, particularly nudge acceptance. We identify health-promoting collaboration and, in particular, the responsibility to promote healthy eating as influential factors for nudge acceptance. Even though these factors still need to be further researched, they provide useful information for understanding nudge acceptance and should be considered in Hagman’s nudge acceptance model [25]. In addition, we suggest to consider the integration of nudging into the HPU framework to especially enhance efforts made for creating healthy work and study environments, as health-promoting collaboration was found to increase nudge acceptance. Our systematic grouping of individuals according to their nudge acceptance further underlines the importance of the source of a nudge—an institution, family, friends, peers, fellow students, or oneself. The possibility of self-initiated nudging also needs to be considered further.

## 5. Conclusions

We found that students have a moderate overall nudge acceptance level, and that nudge acceptance varies from nudge to nudge. The specific design of a nudge thus plays an important role, and a one-nudge-fits-all design should indeed no longer be applied. The perceived responsibility of a cafeteria or university to promote healthy eating, as well as health-promoting collaboration, positively influenced overall nudge acceptance. We identified three clusters of students differing in their nudge acceptance—nudgeable students, conditionally mixed nudgeable students, and un-nudgeable students. These insights offer practical implications for nudge development by university, cafeteria, and public health officials. Officials need to take seriously their responsibility for promoting healthy eating and an environment in which students can support each other regarding their shared healthy eating beliefs and values. In this way, acceptance of healthy eating nudges and their consequent effectiveness can be increased. The nudgeable students are very likely to be positively affected by healthy eating nudges. The conditionally mixed nudgeable students are likely to accept nudges as well when these nudges rely on certain mechanisms (default, salience, priming, and affect) and portray social norms and health-promoting collaboration. For the un-nudgeable students, a self-initiated nudge focusing on default options, salience, and priming is likely to be accepted, at least moderately. These types of nudges, when presented as self-nudging, should allow them to consciously and voluntarily choose when to be nudged. Our findings provide further insights into the theoretical background of nudge acceptance and susceptibility to nudging.

## Figures and Tables

**Table 1 ijerph-19-04107-t001:** Descriptive statistics of nudge acceptance.

Nudge Acceptance	*M* (*SD*)	Cronbach’s α
Overall nudge acceptance scale	2.87 (0.72)	0.741 (9)
Messenger nudge	3.43 (1.35)	-
Incentive nudge 1	3.63 (1.37)	-
Incentive nudge 2	3.45 (1.39)	-
Norms nudge	3.57 (1.39)	-
Default nudge	1.79 (1.14)	-
Salience nudge	1.89 (1.10)	-
Priming nudge	1.86 (1.14)	-
Affect nudge	2.28 (1.21)	-
Priming and salience nudge	3.96 (1.27)	-

All items were measured on a scale from 1 to 5, with 1 indicating higher values; standard deviations and number of items in parentheses; *n* = 1031.

**Table 2 ijerph-19-04107-t002:** Descriptive statistics of the influential factors.

Factor	*M* (*SD*)	Cronbach’s α
Social norms ^1^	2.92 (0.83)	0.610 (3)
Responsibility ^1^	3.50 (1.20)	-
Health-promoting collaboration ^2^	3.17 (0.62)	0.641 (6)
Procrastination ^2^	2.78 (1.17)	0.925 (4)

^1^ Scale from 1 to 5, with 1 indicating higher values; ^2^ scale from 1 to 5, with 5 indicating higher values; standard deviations and number of items in parentheses.

**Table 3 ijerph-19-04107-t003:** Multiple regression of the overall nudge acceptance.

Influential Factor	*b*	*ß*	*p*	*R* ^2^
Constant	2.31 (0.17)		<0.001	0.169
Social norms ^1^	0.05 (0.03)	0.06	0.053	
Health-promoting collaboration ^2^	−0.11 (0.03)	−0.10	<0.001	
Responsibility ^1^	0.23 (0.02)	0.38	<0.001	
Procrastination ^2^	−0.01 (0.02)	−0.01	0.700	

^1^ Scale from 1 to 5, with 1 indicating higher values; ^2^ scale from 1 to 5, with 5 indicating higher values; standard errors in parentheses.

**Table 4 ijerph-19-04107-t004:** Multiple regression analyses results of influential factors on the acceptance of each nudge.

Influential Factor	Nudge								
	1	2	3	4	5	6	7	8	9
Social norms	-	0.10 *	-	-	-	-	-	-	-
Health-promoting collaboration	-	-	−0.15 *	-	−0.14 *	−0.26 ***	-	-	-
Responsibility	0.22 ***	0.20 ***	0.31 ***	0.32 ***	0.23 ***	0.13 ***	0.21 ***	0.26 ***	0.17 ***
Procrastination	-	-	-	-	-	-	-	-	−0.07 *
*R* ^2^	0.050	0.042	0.090	0.092	0.064	0.055	0.062	0.077	0.031

1 = messenger, 2 = incentive 1, 3 = incentive 2, 4 = norms, 5 = default, 6 = salience, 7 = priming, 8 = affect, 9 = priming and salience; * *p* < 0.05, *** *p* < 0.001.

**Table 5 ijerph-19-04107-t005:** Descriptive statistics for the acceptance of the nudge types per cluster.

Type of Nudge	Un-Nudgeable (*n* = 184)	Conditionally Mixed Nudgeable (*n* = 459)	Nudgeable (*n* = 338)
Messenger nudge	1.55 (0.84)	2.40 (1.21)	3.37 (1.28)
Incentive nudge 1	1.61 (0.99)	1.87 (1.03)	3.44 (1.27)
Incentive nudge 2	1.69 (0.92)	2.09 (1.09)	3.58 (1.15)
Norms nudge	1.77 (1.04)	1.78 (0.96)	3.66 (1.19)
Priming and salience nudge	1.47 (0.85)	1.62 (0.93)	2.92 (1.38)
Default nudge	3.22 (1.45)	4.47 (0.87)	4.42 (0.94)
Salience nudge	2.98 (1.24)	4.30 (0.88)	4.49 (0.83)
Priming nudge	2.80 (1.37)	4.43 (0.76)	4.48 (0.81)
Affect nudge	2.50 (1.18)	3.83 (1.07)	4.26 (0.89)

**Table 6 ijerph-19-04107-t006:** Demographic information on the clusters.

		Un-Nudgeable (*n* = 184)	Conditionally Mixed Nudgeable (*n* = 459)	Nudgeable (*n* = 338)
Gender	Female	98 (53.3%)	306 (66.7%)	184 (54.4%)
	Male	84 (45.7%)	147 (32.0%)	152 (45.0%)
	Diverse	2 (1.1%)	5 (1.1%)	1 (0.3%)
Age	17–24 years	120 (65.2%)	341 (74.3%)	240 (71.0%)
	18–30 years	48 (26.1%)	95 (20.7%)	75 (22.2%)
	31 years/older	16 (8.7%)	21 (4.6%)	23 (6.8%)

**Table 7 ijerph-19-04107-t007:** Descriptive statistics for the three clusters, as well as *F*-values comparing these statistics.

Construct	Un-Nudgeable (*n* = 184)	Conditionally Mixed Nudgeable (*n* = 459)	Nudgeable (*n* = 338)	*F* (2, 957)
Social norms ^1^	2.87 (0.82)	3.09 (0.84)	3.18 (0.81)	8.70, *p* = < 0.001
Health-promoting collaboration ^1^	3.05 (0.69)	3.18 (0.60)	3.24 (0.61)	5.32, *p* = 0.005
Responsibility ^1^	1.82 (0.96)	2.41 (1.11)	3.02 (1.20)	70.54, *p* = < 0.001
Procrastination ^1^	2.83 (1.20)	2.74 (1.18)	2.83 (1.13)	0.71, *p* = 0.493

^1^ Scale from 1 to 5, with 5 indicating higher values; standard deviations in parentheses.

## Data Availability

The data presented in this study are available on request from the corresponding author. The data are not publicly available because they were collected in a public institution (university).

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
