# Peer review of "Are You “Nudgeable”? Factors Affecting the Acceptance of Healthy Eating Nudges in a Cafeteria Setting"

_ijerph, 2022, doi:10.3390/ijerph19074107_

Round 1
Reviewer 1 Report
This is a very good quality manuscript. The research problem is relevant and important. The applied methodology is correct. A large sample of respondents was used. The results are presented clearly. The implications are discussed in the right way.
Measuring actual consumer behaviour regarding nudge acceptance for various types of nudges (a field experiment) would be a promising avenue for your future research, which could lead to a publication in a top journal.
line 313 - why no values are provided?
Incomplete references: 1, 6, 7, 16, 25, 30, 32, 36, 40
Author Response
"Please see the attachment."

Reviewer 2 Report
The paper entitled “Are you “nudgeable”? Factors affecting the acceptance of healthy eating nudges in a cafeteria setting” investigates the acceptance of nine healthy eating nudges and explores whether four different factors can influence nudgeability.
The article is based on a well prepared and executed research. The analysis and calculations are well executed, utilizing the relevant statistical methodology. The presentation of the results is good, and discussions of results is quite robust. Overall, the article is an interesting read.
Please find below some minor comments/suggestions:
- Line 74: the authors might want to clarify that the percentage mentioned in this sentence refers to the German general population.
- Lines 257-266: those lines are a bit confusing for the reader. Does the MINDSPACE framework summarize 9 mechanisms or 10? Is there any difference between the MINDSPACE framework and Nørnberg’s scale presented in table A1 in the Appendix?
- Line 270: why did you decide to add the “priming and salience nudge” since the salience nudge and priming nudge are already included separately in the scale?
- Table A1_Appendix: I would suggest the authors to translate their own item (priming and salience nudge) into English. This might help the reader to understand the necessity of this nudge.
- Be consistent with the scale measurement: line 274 – do not agree, though, line 725 – disagree
- I would suggest the authors to include the forms of the model specifications used to estimate results in tables 3 & 4.
Author Response
"Please see the attachment."

Reviewer 3 Report
The topic of the manuscript is interesting and relevant in its field of investigation. It is based on a well-prepared end executed research.
In the introduction, add the structure of the paper. Briefly explain what is in each section.
Researchers recognize the limitations of their research and write about them in section 4.3. Limitations and priorities for future research. It is commendable. This fact proves the high reliability and scientific maturity of the Authors of the presented article.
The references are good and closely related to the topic of the article.
I would like to draw their attention to three articles that can possibly enrich the discussion of results.
- Barker, H.; Shaw, P.J.; Richards, B.; Clegg, Z.; Smith, D. What Nudge Techniques Work for Food Waste Behaviour Change at the Consumer Level? A Systematic Review. Sustainability 2021, 13, 11099. https://doi.org/10.3390/su131911099
- Carroll, K.A.; Samek, A.; Zepeda, L. Food bundling as a health nudge: Investigating consumer fruit and vegetable selection using behavioral economics. Appetite 2018,
- Wilson, A.L.; Buckley, E.; Buckley, J.D.; Bogomolova, S. Nudging healthier food and beverage choices through salience and priming. Evidence from a systematic review. Food Qual. Prefer. 2016, 51, 47–64.
Finally, one more little thing. I suggest mentioning the sources of the presented data under the tables and figures.
Author Response
"Please see the attachment."
